# Development and testing of the 'Culture of Care Barometer' (CoCB) in healthcare organisations: a mixed methods study

Anne Marie Rafferty,[1] Julia Philippou,[1] Joanne M Fitzpatrick,[1] Geoff Pike,[2] Jane Ball[3]

[1]Florence Nightingale Faculty of Nursing and Midwifery, King's College London, London, UK
[2]Employment Research, Hove, UK
[3]Faculty of Health Sciences, University of Southampton, Southampton, UK

**Correspondence to**
Dr Anne Marie Rafferty;
anne_marie.rafferty@kcl.ac.uk

## ABSTRACT

**Objective** Concerns about care quality have prompted calls to create workplace cultures conducive to high-quality, safe and compassionate care and to provide a supportive environment in which staff can operate effectively. How healthcare organisations assess their culture of care is an important first step in creating such cultures. This article reports on the development and validation of a tool, the Culture of Care Barometer, designed to assess perceptions of a caring culture among healthcare workers preliminary to culture change.

**Design/setting/participants** An exploratory mixed methods study designed to develop and test the validity of a tool to measure 'culture of care' through focus groups and questionnaires. Questionnaire development was facilitated through: a literature review, experts generating items of interest and focus group discussions with healthcare staff across specialities, roles and seniority within three types of public healthcare organisations in the UK. The tool was designed to be multiprofessional and pilot tested with a sample of 467 nurses and healthcare support workers in acute care and then validated with a sample of 1698 staff working across acute, mental health and community services in England. Exploratory factor analysis was used to identify dimensions underlying the Barometer.

**Results** Psychometric testing resulted in the development of a 30-item questionnaire linked to four domains with retained items loading to four factors: organisational values ($\alpha$=0.93, valid n=1568, M=3.7), team support ($\alpha$=0.93, valid n=1557, M=3.2), relationships with colleagues ($\alpha$=0.84, valid n=1617, M=4.0) and job constraints ($\alpha$=0.70, valid n=1616, M=3.3).

**Conclusions** The study developed a valid and reliable instrument with which to gauge the different attributes of care culture perceived by healthcare staff with potential for organisational benchmarking.

## Strengths and limitations of this study

► This study explored the reliability, content validity and factorial structure of the Culture of Care Barometer (CoCB) tool.
► The tool was tested with a total sample of over 2000 healthcare staff.
► The tool was only tested in public healthcare settings in England and hence requires further validation across different care settings to evaluate its usefulness and acceptability with wider populations.
► The CoCB requires further validation to ascertain its concurrent and predictive validity.

Care Quality Commission has drawn attention to 'cultures of care that are too often 'task-based' when they should be person-centred, and where the unacceptable become the norm'[5] (p. 5) noting variation in cultures within organisations reflecting leadership and management failings.[6] Learning from high-profile failures in care delivery indicates that quality and culture are not uniform within, let alone across, organisations.[6 7] This was evident in the description and analysis of events (and the context to those events) at Mid Staffordshire National Health Service (NHS) Trust, described by the Robert Francis Inquiry.[1] Pockets of excellence can coexist alongside the worst examples of care failures[6]; lack of consistency in care culture impedes the spread of good practice across organisations.[1 8] Establishing cultures that will allow healthcare organisations to achieve the ultimate goal of providing high-quality care has therefore become a major policy concern.

Evidence suggests that major failures are frequently not brought to light by the systems for quality assurance or improvement that are part of most healthcare organisations in developed countries, such as incidence reporting, mortality and morbidity reviews, inspections, accreditations, clinical profiling, and risk and claim management.[3] Since these cultural

## BACKGROUND

The importance of culture in providing high-quality and safe care to patients has been emphasised in many investigations of failings in healthcare systems both nationally[1 2] and internationally.[3] Healthcare organisations have begun to look critically at ways that can improve their culture and consequently the care provided to patients.[4] In the UK, the

attributes are not picked up in the measures of quality and performance currently in use, metrics fail to capture the meaning and reality of a culture of care for patients or staff. Moreover, research in the UK demonstrates that the well-being of staff is closely linked to the well-being of patients, and staff engagement is a key predictor of a wide range of outcomes in healthcare organisations.[9]

One of the first questions organisations have to consider in trying to establish such care cultures is how to assess the organisational culture. The first difficulty in this is that the concept of culture is broad and multifaceted.[10] While culture as a concept is widely used, the term itself has been described as an 'indescribable mist'.[10] Conceptual debates over how culture is defined continue and consequently impact how it is studied.[10] Much of the literature in healthcare favours the concept of culture as shared beliefs, norms and routines through which a society can be interpreted and understood.[11] With this definition in mind, our focus was on understanding the culture of care, as a subset of organisational culture, and helping organisations gauge the different attributes of caring environments.

From an organisational development (OD) perspective, the practical application of culture assessment tools speaks to the diagnostic premises of OD theory and practices. Such tools ascertain the strengths and weaknesses of organisations and help them prescribe interventions or 'treatments' of change based on an objective diagnosis from the data collected.[12] Within healthcare, there is a plethora of well-known instruments for measuring the culture of organisations and 'patient safety' culture.[13] A national survey of healthcare organisations in the UK to identify the culture assessment tools that are used within the English NHS concluded that while organisations are increasingly using culture assessment instruments these focus primarily on the assessment of safety culture rather than perspectives of quality.[13] Moreover, while the centrality of patients' experiences, of safety, caring and supportive cultures, in such tools is well evidenced, a large research programme examining culture and behaviour in the English NHS concluded that creating caring cultures where staff can feel supported, respected, valued and engaged are equally important for providing high-quality care.[7] Therefore, it is argued that achieving the optimal care culture is only possible in organisations where staff feel valued, respected and supported and when relationships are good between managers, staff, teams, departments and across institutional boundaries.[7 9 10] An initial analysis of the literature revealed a lack of instruments for measuring 'care cultures' from the perspective of service providers as distinct from organisational culture or patient safety culture.[14]

Diagnostic OD entails a problem-based approach where organisations are considered to be problematic and need fixing.[15] This approach to data gathering has been described as 'problem-sensing'[7] as it actively seeks out weaknesses in organisational systems. This can result in organisational members being wary as they may feel that the main purpose of data collection is to attribute blame; ultimately, this can inhibit or make members more resistant to change.[7 12] One of the most sensitive messages coming out from the Mid Staffordshire Inquiry was how staff suffered as a result of raising concerns.[1] This has led to a review of the way NHS organisations deal with concerns raised by NHS staff, advocating for a culture of safety and learning in which staff feel safe to raise concerns and these conversations take place as part of everyday practice without fear of blame or recrimination.[15] Previous research identified that extant culture assessment tools failed to address culture attributes that promote the development of a blame-free environment.[13]

In contrast to diagnostic OD approaches, dialogical approaches are primarily concerned with 'meaning-making'.[12] Building a more dialogical approach to OD,[12] therefore, could encourage reflection and stimulate discussion about the culture of an organisation and how 'caregivers' express and create meaning in their performance of care.[14] Given the gaps in current culture measurement tools, we aimed to develop a tool that could act as a 'diagnostic' measurement to help organisations assess the culture of care but also as a 'dialogic' tool designed to prompt reflection on the underlying issues involved in creating a caring culture. The current paper presents the development and testing of the Culture of Care Barometer (CoCB) as a tool that has the potential to serve these purposes.

## METHODS

We followed well-recognised and comprehensive approaches for instrument development and testing, for example, Hinkin's framework for scale development,[16–19] and pursued a variety of data collection and analysis methods to operationalise the elements that are important in creating caring cultures and to ensure the reliability of the tool.[20] A detailed account of the process of developing and testing the CoCB is provided below. Figure 1 provides an overview and graphical representation of this process.

### Item generation

We used a mixed method approach in the creation of items to assess the culture of care construct. Our processes involved both inductive and deductive approaches to generate items.[16–18] Initially, an inductive approach was used where an expert panel of six healthcare leaders developed a prototype questionnaire by generating items and domains they considered important in improving patient care. The expert panel consisted of individuals with extensive experience and expertise in regulation, leadership, healthcare delivery, management, policy and research within the English NHS, including the use of tools used in inspection regimens. This process was complemented by a deductive approach involving a comprehensive literature review in four major healthcare-related electronic database resources (CINAHL, Embase, MEDLINE and

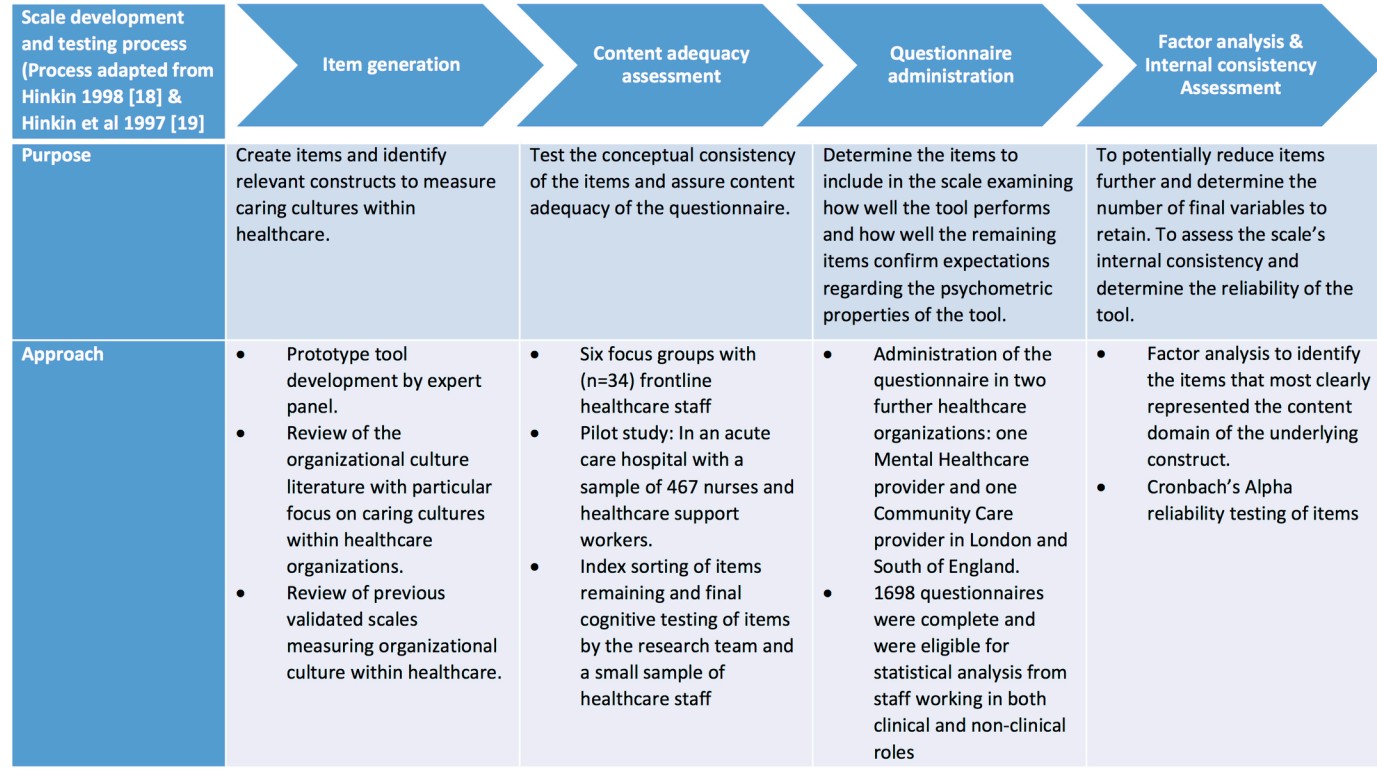

| Scale development and testing process (Process adapted from Hinkin 1998 [18] & Hinkin et al 1997 [19] | Item generation | Content adequacy assessment | Questionnaire administration | Factor analysis & Internal consistency Assessment |
|---|---|---|---|---|
| **Purpose** | Create items and identify relevant constructs to measure caring cultures within healthcare. | Test the conceptual consistency of the items and assure content adequacy of the questionnaire. | Determine the items to include in the scale examining how well the tool performs and how well the remaining items confirm expectations regarding the psychometric properties of the tool. | To potentially reduce items further and determine the number of final variables to retain. To assess the scale's internal consistency and determine the reliability of the tool. |
| **Approach** | • Prototype tool development by expert panel.<br>• Review of the organizational culture literature with particular focus on caring cultures within healthcare organizations.<br>• Review of previous validated scales measuring organizational culture within healthcare. | • Six focus groups with (n=34) frontline healthcare staff<br>• Pilot study: In an acute care hospital with a sample of 467 nurses and healthcare support workers.<br>• Index sorting of items remaining and final cognitive testing of items by the research team and a small sample of healthcare staff | • Administration of the questionnaire in two further healthcare organizations: one Mental Healthcare provider and one Community Care provider in London and South of England.<br>• 1698 questionnaires were complete and were eligible for statistical analysis from staff working in both clinical and non-clinical roles | • Factor analysis to identify the items that most clearly represented the content domain of the underlying construct.<br>• Cronbach's Alpha reliability testing of items |

**Development and testing of the Culture of Care Barometer (CoCB) Tool**

**Figure 1** Process of developing and testing the Culture of Care Barometer tool.

Web of Science, from 1945 to 2015) using key terms such as 'organisational culture', 'assessment tools', 'healthcare settings' and 'quality of care'. Through the literature review, key studies[7 21–24] and previous comprehensive reviews on validated scales measuring organisational culture[13 14 25 26] within healthcare were extracted and together these two complementary processes helped us create an initial list of candidate items and facilitated the refinement of items and elaboration of domains that could be used to measure the concept of the culture of care.

## Content adequacy assessment

A comprehensive approach was also followed at the stage of content adequacy to test the conceptual consistency of the items and assure content validity of the questionnaire. To achieve this, we undertook six focus groups with a cross-section of healthcare staff (n=34) of different levels of seniority and from different settings, for example, both inpatient and outpatient. The aim of these focus groups was to explore perceptions of terminology and cognitively test items and their meaning to enable a 'co-creation approach' to the development of the tool with frontline staff. This was envisaged as a process that would allow the tool development stage to address some limitations reported in the literature in terms of developing tools that are 'fit for purpose' in the NHS context. The process also allowed us to explore staff ideas of what constitutes a 'good culture of care' and key 'signs and symptoms' of

an organisation that has a good or poor culture of care and identify any additional items relevant to the concept. At the same time, we were able to test the appropriateness, comprehension and clarity of items, and ease and acceptability of instructions and format of the initial questionnaire.[20]

At this stage, an initial tool of 37 items clustered under four domains was developed, and a 5-point scale, ranging from not at all to fully agree, was used to record participants' agreement with the items. The first domain comprised six items that explored issues surrounding resources and quality of care, facilities and equipment, staffing levels and views of workplace in terms of safety and quality. It included the Friends and Family Test as a reference item[27] and a question about action required to improve resources. The second domain comprised 10 items relating to management and support. Ten items in the third domain addressed development, staff involvement in decision making and overall culture in the organisation. The fourth domain included 11 items about staff meetings, teamwork and feedback and willingness of the organisation to learn from issues raised as well as incidents. For each domain, a question was included about how much influence participants had to improve things, rated on a 5-point scale (from 'none' to 'a lot'). The initial tool is provided in full in the study's report.[28]

The tool was then pilot tested in an acute NHS organisation with a sample of registered nurses and midwives and

healthcare support workers (n=467, 24% response rate). The aim of this pilot testing was to examine how well the tool performed and test the face validity and internal reliability of the items before wider testing. An initial factor analysis was conducted to reduce the number of items and further refine the tool. Results from this pilot study are reported in the study's report.[28] The items identified through this analysis were explored by the research team using an index-card-sorting exercise.[29] This involved asking the members of the research team to sort the items into categories. The items were printed separately on a small index card, each member of the team sorting the cards into groups. Each then described what they saw as the common theme relating to the cards in each group. Through this analysis and process, 30 items were retained in the questionnaire representing the concept of culture of care, and these were classified under seven categories: engagement, empowerment, management and leadership, values, roles, resources and team. Participants were asked to rate the extent to which they agreed with each statement on a 5-point Likert scale (from strongly disagree to strongly agree). Before administering the final questionnaire to a wider sample, a small number of NHS staff reviewed the items and the overall questionnaire to identify whether this provided an accurate representation of the overall culture in their workplace. The revised version of the tool is available in full in the study report.[28]

### Questionnaire administration

The retained items and revised tool were administered to a wider sample with the objective of examining how well the tool performed and how well the remaining items confirmed expectations of the psychometric properties of the tool. Below we present the main procedures we followed during this stage of development and testing the CoCB tool.

Administration of the questionnaire was undertaken in two further healthcare organisations. One was a mixed mental healthcare provider in London with community care and inpatient beds, and the other was a predominately community healthcare organisation in a mixed urban/rural setting in the South of England. A total sample approach was adopted, and an electronic version of the questionnaire was sent to the two organisations for distribution. We also forwarded 1500 paper copy questionnaires for each organisation to distribute to participants with less access to the electronic version. Questionnaires were distributed in June 2014 and data collection lasted for about 8 weeks.

### Factor analysis and internal consistency assessment of the tool

Data were coded and entered into a Microsoft Excel spreadsheet before being exported and analysed through IBM SPSS V.21 statistical software. The demographic and employment characteristics of the participants were analysed descriptively and presented as numbers and percentages. As the measurement model might differ

between types of organisations (eg, acute, community and mental health) and the type of sample (eg, clinical and non-clinical staff), an exploratory factor analysis was undertaken to identify whether the correlations between groups of observed items originated from one or more latent variables/factors in the data. Internal consistency was tested using Cronbach's coefficient α. An alpha reliability score above 0.7 is considered good[19 29]; therefore, the target level of reliability was set at 0.70.

## RESULTS

### Sample characteristics

A total of 1705 staff working across mental health and community services in both clinical and non-clinical roles returned the questionnaires, organisation 1 (n=700, 25% response rate); organisation 2 (n=1005, 24% response rate), of which 1698 were found complete and eligible for statistical analysis. Table 1 summarises the main profile and employment characteristics of participants. Overall, four in five (82%, n=1237) respondents were female, with a large proportion of participants indicating being aged 40 years or older (table 1). Almost all sample participants (93%, n=1459) spoke English as their first or main language. There were no difference in the demographic variable by organisation. Two-thirds of staff across the two organisations worked full-time (68%, n=1074). By staff group, there was a similar response profile within the two organisations, and overall a larger proportion of nurses responded to the questionnaire compared with other groups of staff.

### Structure of the tool

A factor analysis was performed on the 30 items to identify patterns of loading and extract underlying factors. Through this analysis, four factors were identified. The composition of the factors was based on aggregating the scores for each item to create a single score for each factor and dividing this figure by the number of items in the CoCB tool to provide a meaningful average score. The relevant loading of items for each of the four factors is presented in table 2.

The 12 items of the first subscale were predominately related to the macro-level elements within organisations that influence culture such as valuing employees, good communication within the organisation and visible leadership at the top level. Therefore, this subscale was considered to address wider 'organisational values'. Eleven items were loading to the second factor labelled as team support. Items loading to this subscale were concerned with the 'meso level' of organisations and included elements that described primarily team support relationships and management and development of employees within organisations. The remaining seven items concerned aspects of everyday work at the micro level within organisations and these were loading to two factors. One subscale was mainly associated with four items describing social elements of work such as respect

| Table 1 | Characteristics of participants | | |
|---|---|---|---|
| | Percentage (n) | | |
| | Organisation 1 (n=700) | Organisation 2 (n=1005) | Overall (n=1705) |
| Female | 75 | 87 | 82 |
| English as a first or main language | 86 | 98 | 93 |
| Age groups, years | | | |
| <29 | 8 | 8 | 8 |
| 30–39 | 17 | 17 | 17 |
| 40–49 | 33 | 32 | 32 |
| 50–59 | 32 | 35 | 34 |
| 60 plus | 9 | 8 | 8 |
| Staff group | | | |
| Registered nurses and health visiting staff | 30 | 34 | 32 |
| Healthcare assistant/support worker | 13 | 5 | 8 |
| Allied health professionals | 25 | 25 | 25 |
| Estates and facilities | 2 | 4 | 3 |
| Doctor/dentist | 5 | 3 | 4 |
| Administrative and clerical staff | 15 | 15 | 15 |
| Central functions and corporate services | 4 | 6 | 5 |
| Other | 4 | 6 | 5 |
| Work setting | | | |
| Community | 36 | 45 | 41 |
| Clinics/outpatient departments | 13 | 14 | 14 |
| Wards/inpatient units | 27 | 7 | 15 |
| Office | 16 | 27 | 23 |
| Other | 9 | 7 | 7 |
| Employment status | | | |
| Full-time | 76 | 64 | 68 |
| Part-time | 25 | 36 | 32 |

and social support between coworkers, and the final three items were concerned with the ability of employees to do their job within the limits of time and resources available to them (factor 4).

Reliability analyses were performed on each factor to identify how items loading to factors were considered a positive endorsement of the subscale. The results of these analyses are presented in table 3. Cronbach's α for the macro-level (factor 1: organisational values) and meso-level scales (factor 2: team support) were very high both at 0.93. Cronbach's α for the micro level scales were 0.84 (factor 3: relationships with colleagues) and 0.70 (factor 4: job constraints).

Both sorting and factor analytical techniques were used to assess the content adequacy of the 30 items. The initial sorting of the items undertaken as part of the content adequacy process identified seven themes. The exploratory factor analysis did not confirm the distinction among the seven themes and instead the results yielded a four-factor solution with factors indicating greater emphasis for organisational values and team and social

support compared with job constraints. Although this process of developing the tool provides confidence in the four factors identified, confirmatory factor analysis with another independent sample will provide a more rigorous test of the loading of items.

## Usefulness and added value of the tool to healthcare organisations

The results from the CoCB were presented to the participating healthcare organisations through two independent reports detailing the findings from testing the tool in each organisation. Following the presentation of the results, we invited key individuals from each organisation (n=5) to a follow-up discussion to receive feedback on the usefulness of the tool and explore whether the tool met the dialogical and diagnostic premises of OD.[12] These individuals held strategic leadership positions within the two organisations with responsibilities for overseeing workforce and culture initiatives. These sessions were audio recorded and analysed thematically.

**Table 2** Factor analysis and loading of items

| Subscales and items | Loading |
| --- | --- |
| Factor 1: organisational values (macro level) | |
| The Trust listens to staff views | 0.84 |
| The Trust has a positive culture | 0.77 |
| There is strong leadership at the highest level in the Trust | 0.75 |
| I am able to influence how things are done in the Trust | 0.74 |
| I would recommend this Trust as a good place to work | 0.70 |
| I feel well informed about what is happening in the Trust | 0.70 |
| Staff successes are celebrated by the Trust | 0.70 |
| The Trust values the service we provide | 0.69 |
| Trust managers know how things really are | 0.68 |
| I am proud to work in this Trust | 0.65 |
| A positive culture is visible where I work | 0.50 |
| I get the training and development I need | 0.40 |
| Factor 2: team support (meso level) | |
| I feel well supported by my line manager | 0.87 |
| My line manager treats me with respect | 0.84 |
| My line manager gives me constructive feedback | 0.83 |
| My concerns are taken seriously by my line manager | 0.81 |
| I feel part of a well-managed team | 0.60 |
| I am kept well informed about what is going on in our team | 0.52 |
| I feel supported to develop my potential | 0.50 |
| I know who my line manager is | 0.45 |
| I am able to influence the way things are done in my team | 0.44 |
| I feel able to ask for help when I need it | 0.44 |
| Unacceptable behaviour is consistently tackled | 0.40 |
| Factor 3: relationships with colleagues (micro level) | |
| The people I work with are friendly | 0.81 |
| When things get difficult, I can rely on my colleagues | 0.79 |
| I feel respected by my coworkers | 0.76 |
| I have positive role models where I work | 0.56 |
| Factor 4: job constrains (micro level) | |
| I have sufficient time to do my job well | 0.79 |
| I have the resources I need to do a good job | 0.72 |
| I know exactly what is expected of me in my job | 0.41 |

From a diagnostic perspective, the consensus overall was that the CoCB tool resonated with other instruments used in the organisations, adding 'colour and depth' to them.

*I think it is a much richer type of feedback than we get from the staff survey. We liked the logic and flow and could appreciate the sense of questions. (Organisational Development Manager, Organisation 2)*

*…you get these action plans that come out of the staff surveys, but the detail is not the depth of information that we had with this, so this enables you to think more about why, then, and question, as oppose to, 'oh, right, we've got to do something on that area. (Senior Manager, Organisation 1)*

The brevity of the tool, the fact that it was easy to complete and was perceived as targeting the right domains, was appreciated by staff. Moreover, from a diagnostic perspective, the tool was perceived by participants as useful in providing a reference point for them to gauge where they were on a cultural spectrum or journey.

From a dialogical perspective, the fundamental value of the CoCB tool was reflected in the belief that 'culture changes by talking about it' and the Barometer helped to surface issues for discussion. Data from the CoCB were seen as helpful in drilling into further detail or using it as a prompt for a 'quality conversation' for instance, with smaller, discrete groups, teams or where it was felt things were not quite right or when organisations felt the need to gauge the impact of changes they had made.

*Culture …does not change overnight and the fact that it is (the CoCB tool) a prompt to reflect on has been I think a really powerful aspect of the tool. (Senior Executive, Organisation 2)*

Finally, the commentary element of the tool was identified as providing a rich source of intelligence in helping to unpack notions of culture:

*trying to understand what it is that matters to staff and what they feel about the place that they work in. (Workforce Transformation Lead, Organisation 2)*

It was also regarded as helpful in picking up on contradictions that might exist in organisations, as one senior manager observed:

*where you've got high scores for 'my manager treats me with respect', but then, 'Oh, I don't think my manager understands what the real world is like' (Leadership Project Manager, Organisation 1)*

Richer feedback via the CoCB in comparison with the staff survey helped to tease out the contradictions and take the quality of conversations at team level to the next stage.

## DISCUSSION

Enabling the workforce to put the right things in place for patients is key to improving NHS performance in terms of quality and safety,[30 31] and this is the underlying principle of the CoCB as a tool. The challenge all organisations face is that there is not a one-size-fits-all solution as

| Table 3 | Scale characteristics (Cronbach's coefficient α reliability analysis) | | | |
|---|---|---|---|---|
| | Factor 1: organisational values (macro level) | Factor 2: team support (meso level) | Factor 3: relationship with colleagues (micro level) | Factor 4: job constrains (micro level) |
| Total items loading | 12 | 11 | 4 | 3 |
| Alpha reliability | 0.93 | 0.93 | 0.84 | 0.70 |
| Number of valid responses | 1568 | 1557 | 1617 | 1616 |
| Mean score | 3.7 | 3.2 | 4.0 | 3.3 |

each individual is unique and will react differently to the challenges and values of an organisation. The CoCB can provide feedback from staff to enhance understanding of the factors that contribute to a culture lacking in care and safety. The CoCB appeared to perform well in meeting the gap in the literature that suggests that there is a need for tools to assess culture with a focus on formative diagnostic purposes to support reflexive practice.[13] The focus on the carers' views helps to comprehend the intersection of individual and organisational factors that distinguish the CoCB from other tools that prioritise either macro or micro levels. Moreover, using a dialogical approach to OD, we created a tool that can be a resource to facilitate the involvement of frontline staff, at different levels and with different roles, in culture change initiatives.

The CoCB tool possesses several advantages over existing tools. During the development and testing of the tool, we addressed concerns reported in the literature and from participants about the need for tools that are 'fit for purpose'[13] and are not onerous and time consuming to complete. One of the major strengths of this study was the cocreation of the tool in collaboration with frontline staff providing care within NHS organisations. Three main sources of data and information were used to create the content of the tool, namely a prototype tool developed by an expert panel in this field, a comprehensive literature review of the concept of culture of care and other tools used within healthcare, and input from interviews with staff working in healthcare organisations. Feedback from participants indicated that the data collected via the CoCB could provide extra intelligence about the jigsaw of what is happening within organisations. Participants indicated that the tool could enhance or complement current data collection methods used in hospital settings such as the Friends and Family Test and the Staff Survey to identify concerns about poor care. In this way providing a more comprehensive picture of patient and staff satisfaction with services provided. In addition, participants spoke of the struggle to make meaningful changes based on Staff Survey feedback alone, and they shared that data collected via the CoCB alerted them to important contextual feedback and factors that in some cases were more useful in planning and developing action plans.

While the CoCB is a reliable tool and does seem to fill a niche for identifying and understanding some of the social processes at work within an organisation, we recognise that not all factors contributing to developing a caring culture may be included in the tool. We also acknowledge that not all the items may be relevant to all healthcare contexts, and we recommend that testing and adaptation of these items may be necessary in future validations. Therefore, practitioners should use the CoCB in conjunction with other tools that can help organisations achieve culture change. Moreover, the sample of this testing was predominately nurse led, which may be a direct result of the proportion of nurse personnel working in the healthcare environments. However, cultures are cocreated by all members of an organisation, and this means that everyone is responsible for the welfare of the organisation as a whole. Further work is needed to test the reliability and validity of the questionnaire as well as engagement and uptake of the tool with other professional groups. While the tool has been developed in the context of the UK NHS, future work to adjust and test the tool in other organisations and countries will allow validation of the tool in a global context. While the development of the tool was prompted by an interest in the contexts that may support the delivery of patient centric care, we have yet to examine if environments with more positive cultures of caring do indeed have care that is considered to be more patient centric—by staff or patients. Future work could investigate the hypothesised relationships between the culture of care and achievement of patient-centred care delivery, staff satisfaction, work engagement and a reduction in work-related burnout as well as student learning.

Measuring and monitoring culture is a recurrent challenge in healthcare; the CoCB could act as a diagnostic and practical tool for organisations to embrace as a first step in improvement work as well as a means of monitoring change over time. The tool has potential use in clinical practice and research. It is easy to administer, can be completed and analysed quickly providing timely feedback that can be used by organisations to identify areas of strength and weakness and help with the planning of continuous quality improvements or culture change initiatives that hospitals are undertaking. The wider applicability of the tool needs to be explored in future studies and its relevance for groups that may not have a direct role in care provision but who nevertheless are an important part of the organisation and whose feedback is important as culture is every one's business.[23] At the time of writing this paper, a digital version of the tool is being developed. Further research should extend the current efforts to refine and evaluate the impact of the CoCB

as necessary in order to develop interventions that can improve the culture of care in healthcare environments.

**Acknowledgements** We would like to thank all those who contributed to this work, particularly the staff who took part in the survey and those who engaged in the focus groups. We also acknowledge the role of the Passionate About Care Today 'PACT' group (Baroness Audrey Emerton, Professor Dame Elizabeth Fradd, Professor Tricia Hart, Sir Stephen Moss, Flo Panel Coates and Professor Anne Marie Rafferty) in creating the prototype Culture of Care Barometer upon which this work is based and for their time and insights throughout the subsequent development of the tool. We are grateful to the support and insights provided by the Steering Group for the project: Caroline Alexander, Bronagh Scott, Flo Panel Coates, Yvonne Coghill, Morvia Gooden, Sylvia Tang, Virginia Minogue, Jane Clegg, Paul Taylor, Nigel Charlesworth, Hana Wild and Savaia Stevenson.

**Contributors** AMR and JP are joint first authors. AMR was the principal investigator and with JB conceived and designed the study. All authors contributed to the collection, analysis or interpretation of data for the work. JP developed the first draft of the paper, and all the authors contributed to different sections and revised the paper critically for intellectual content. All authors gave final approval of the version to be published and agreed to be accountable for all aspects of the work in ensuring that questions related to accuracy or integrity of any part of the work are appropriately investigated and resolved.

**Funding** This work was supported by funding received from NHS London and NHS England. The independent research reported in this paper formed part of the Compassion in Practice, National Strategy for Nursing, Midwifery and Care Staff, led by Jane Cummings, the Chief Nursing Officer for NHS England, under Action Area 4 'Building and Strengthening Leadership-leading with Compassion'.

**Competing interests** None declared.

**Ethics approval** The project has been approved by the Psychiatry, Nursing and Midwifery Research Ethics Subcommittee (PNM RESC) at King's College London (project no. PNM/13/14-153) and relevant Research and Development (R&D) departments of participating organisations.

**Provenance and peer review** Not commissioned; externally peer reviewed.

**Data sharing statement** Other data relating to the development of the CoCB tool are available by emailing AMR, anne_marie.rafferty@kcl.ac.uk.

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
