## [Reviewer comments · BMJ Open]

ARTICLE DETAILS

TITLE (PROVISIONAL)	Development and testing of the 'Culture of Care Barometer' (CoCB) in healthcare organisations: a mixed-method study.
AUTHORS	Rafferty, Anne Marie; Philippou, Julia; Fitzpatrick, Joanne; Pike, Geoff; Ball, Jane

VERSION 1 - REVIEW

REVIEWER	Moshe Frenkel MD Institute of Oncology Meir Medical Center Israel
REVIEW RETURNED	01-Apr-2017

GENERAL COMMENTS	Well written manuscript with an excellent idea and implementation. A few minor comments: The questionnaire is not clear as to the scale used for each item in the questionnaire, was it a yes - no answers or a sub-scale graded at 0-10 etc. The questionnaire will need to be adjusted to fit organizations outside the NHS, but... it can be a base for a global tool tested in other organizations and countries. It would be interesting to observe if the implementation of this questionnaire will lead health care organizations to be more patient centered and improved healthcare provider satisfaction with reduced burnout. (Authors might relate to this point in their discussion)
--

REVIEWER	Beth Lown The Schwartz Center for Compassionate Healthcare Boston, MA, USA
REVIEW RETURNED	03-Apr-2017

GENERAL COMMENTS	Many thanks to the authors for creating this much needed instrument to assess cultures of care. The text and figure describe very well the dialogic, participatory process the authors used to create the instrument and stress its use to provide formative feedback and prompt dialogue and reflection in the field. The subscales and items are appropriate and well considered. It will be interesting to explore in subsequent research how the CoCB tool correlates with work engagement and conversely, work-related burnout. I am not a statistics expert, but I know from past work that psychometricians have variously applied Mokken analysis and Rasch analysis to scales I have developed. I wonder if additional statistical review would be helpful to the authors.
---

	I look forward to seeing this manuscript in print.
--	--

REVIEWER	Paul Slater Institute of Nursing and Health Research Ulster University Newtownabbey Belfast Northern Ireland
REVIEW RETURNED	11-May-2017

GENERAL COMMENTS	A very interesting and well written paper. The structure of the paper is well set out. You simplified a very complicated process down and still maintained the important message. I really enjoyed reading this paper and well done. Minor corrections/suggestions aimed at strengthening the paper. Page 5 – item generation: How many items were generated in the initial pool? Did it change the domains covered or supplement them? Could you include a KMO test for appropriateness of performing a factor analysis at both quantitative time points as this would strengthen the argument for factor analysis. Include the respondent: item ratio at both survey time points and a reference as why this is important. For example survey two = 56.6:1 ratio. You move from a 4 domain structure to a 7 domain structure following item reduction and sorting, following the EFA. Especially as you move back to 4 factors following the second factor analysis – how does this differ from your original 4 factor model. Is it that the expert 'sorting' differed considerably from the data driven factor structure? Conceptually, can you explain this movement? Why conduct a principal component EFA on the larger data set, given that you have a measurement model you could test based on your initial factor analysis findings. Page 11 line 24: when making reference to 'post Francis' (Francis report) please ensure that international readers are able to understand the reference. Page 12 line 18 typo at end of sentence. Thank you,
--

VERSION 1 – AUTHOR RESPONSE

Reviewer 1: Thank you for your feedback and points.

A sentence was added to clarify this scale used for each item in the questionnaire: "Participants were asked to rate the extent to which they agreed with each statement on a 5-point Likert scale (from strongly disagree to strongly agree)."

We also added a paragraph indicating future work and potential of the tool to be used in other settings outside the NHS:

"While the tool has been developed in the context of the UK NHS, future work to adjust and test the tool in other organisations and countries will allow validation of the tool in a global context. Whilst the development of the tool was prompted by an interest in the contexts that may support the delivery of patient centric care, we have yet to examine if environments with more positive cultures of caring do indeed have care that is considered to be more patient centric – by staff or patients. Future work

could investigate the hypothesized relationships between the culture of care and achievement of patient-centred care delivery, staff satisfaction, work-engagement and a reduction in work-related burnout as well as student learning."

Reviewer 2: Thank you for your feedback and points that enabled us to elaborate further on future work that can be undertaken. This was a point raised from the first reviewer as well and we added the following paragraph in our discussion: "While the tool has been developed in the context of the UK NHS, future work to adjust and test the tool in other organisations and countries will allow validation of the tool in a global context. Whilst the development of the tool was prompted by an interest in the contexts that may support the delivery of patient centric care, we have yet to examine if environments with more positive cultures of caring do indeed have care that is considered to be more patient centric – by staff or patients. Future work could investigate the hypothesized relationships between the culture of care and achievement of patient-centred care delivery, staff satisfaction, work-engagement and a reduction in work-related burnout as well as student learning."

Mokken and Rasch analysis: Given the uncertainty around the dimensionality of the data, Mokken and Rasch analysis seemed not appropriate due to its unidimensional approach that assumes only one latent variable is being measured. Both Rasch and Mokken seem to be about testing people's abilities and this was not the focus of the Culture of Care Barometer tool.

Reviewer 3: Thank you for your feedback and points.

Both sorting and factor analytical techniques were used to assess the content adequacy of the 30 items. The initial sorting of the items undertaken as part of the content adequacy process identified seven themes. The exploratory factor analysis did not confirm the distinction among the seven themes and instead the results yield a four factor solution with factors indicating greater emphasis to organisational values and team and social support compared to job constraints. While this process of developing the tool provides confidence to the four factors identified, confirmatory factor analysis with another independent sample will provide a more rigorous test of the loading of items. We added this paragraph in the manuscript, under section structure of the tool, to clarify the process.

The first sample was used as a pilot phase and only involved nurses and healthcare support workers in one acute organisation. This was initially to help us develop an initial tool and test the appetite of such tool in the healthcare settings. Once this was tested, and since Culture is more inclusive, for the final development of the tool we approached another sample from community and mental health setting including both clinical and non-clinical staff. We assume that the measurement model might differ between types of organisations (e.g. acute, community and mental health) and the type of sample (e.g. clinical and non-clinical staff) and that was the reason for conducting another EFA on a much larger sample. Ultimately we the online tool becoming available soon we want to collect a third sample with data from acute, community and mental health organisations to test the fit of the four factor model, and also to see whether the measurement model is invariant across organisations and type of staff.

We added a sentence to clarify this point under section Factor analysis and internal consistency of the tool: "As the measurement model might differ between types of organisations (e.g. acute, community and mental health) and the type of sample (e.g. clinical and non-clinical staff); an exploratory factor analysis was undertaken to identify whether the correlations between groups of observed items originated from one or more latent variables/factors in the data."

Thank you for all your comments and feedback that helped us strengthen our paper.